# Metabolite Profile and Immunomodulatory Properties of Bellflower Root Vinegar Produced Using *Acetobacter pasteurianus* A11-2

**DOI:** 10.3390/foods9081063

**Published:** 2020-08-05

**Authors:** Na-Young Gil, Hee-Min Gwon, Soo-Hwan Yeo, So-Young Kim

**Affiliations:** Fermented & Processed Food Science Division, Department of Agrofood Resources, National Institute of Agricultural Sciences, Rural Development Administration (RDA), Wanju 55365, Korea; gilnayoung@korea.kr (N.-Y.G.); vitamin89@korea.kr (H.-M.G.); yeobio@korea.kr (S.-H.Y.)

**Keywords:** *Acetobacter pasteurianus*, bellflower root, metabolite, vinegar

## Abstract

Fermented vinegar is prepared from grains and medicinal plants. Here, we produced vinegar from peeled and unpeeled roots of bellflowers (*Platycodon grandiflorum*) using *Acetobacter pasteurianus* A11-2 and analyzed bellflower vinegar (BV) samples using gas chromatography–mass spectrometry and quadrupole time-of-flight mass spectrometry over 15 days of fermentation to assess the quality. We also evaluated their antibacterial and immunoenhancing effects using RAW 264.7 macrophage cells. The major metabolites in BV are organic acids, with the main volatile compounds being ethyl acetate, isoamyl acetate, 1-pentanol, hydroxypropanoic acid, and malonic acid. When we fermented BV from unpeeled roots for 10 days with a starter culture, we observed significant antibacterial and immunoenhancing effects in macrophages. Therefore, we could determine the metabolite and functional differences in vinegar obtained from bellflower roots and proposed that bellflower roots with peel are an effective substrate for developing vinegar and healthy food products.

## 1. Introduction

Vinegar is flavoring agent used worldwide to enhance sour taste and is also considered a medicinal food [1]. Commercial vinegar is classified into two types: synthetic, like glacial acetic acid, and fermented, which is obtained by fermenting alcohols, cereals, and fruits with acetic acid-producing bacteria. Fruit vinegar is made from apples, grapes, persimmons, or pineapples [2,3]. The most widely known type of fruit vinegar is balsamic vinegar, which is mainly used across Europe [4]. In Asia, fermented grain vinegar is made from rice, barley, sorghum, or wheat bran, using starters like nuruk or koji, and brewed by solid-state fermentation [4].

Vinegar is mainly composed of sugars, amino acids, esters, and organic acids, including acetic acid, which is produced by alcoholic and acetic fermentation [5]. Vinegar also contains phenolic compounds and inorganic salts derived mainly from raw materials. These compounds exhibit different biological activities, such as recovery from exhaustion, antiobesity, antitumor, and antioxidant effects [6,7,8]. Unlike phenolic compounds, secondary metabolites are transformed by microorganisms during fermentation [9]. However, most studies on vinegar produced from grains, fruits, or medical plants containing a high concentration of bioactive compounds have rarely explored the correlation between metabolites and bioactive effects in the fermentation of vinegar. Ginseng, *Platycodon grandiflorum,* and *Schisandra chinensis* are considered to be some of the most common herbs used as food.

*P. grandiflorum* (Figure 1) is an annual herb belonging to family Campanulaceae and is used in the preparation of food and drugs in Asia. In addition, the production of vinegar using *P. grandiflorum* has been mentioned in Joseon Dynasty of Korea (1800–1827 A.D.), a well-known ancient document in Korea. The main components of *P. grandiflorum* are amino acids, fibers, vitamins, and other essential trace elements that enhance human health [10]. In addition, *P. grandiflorum* also contain several bioactive compounds, such as flavonoids, phenolic acids, saponins (platycodin A, C, and D and polygalacin D), and sterols (beta-sitosterol and spinasterol) [11,12]. Some of these compounds have anti-inflammatory, antiobesity, antioxidative, antitumor, immunoenhancing, and respiratory-function-enhancing effects [13,14]. The main bioactive components in *P. grandiflorum* are platycodin A, platycodin C, and platycodin D [10,13]. Generally, platycodin D is found in all parts of the plant, with a higher concentration in the lateral parts and root peels than in the aerial parts [10]. *P. grandiflorum* have significant positive effects that allow them to be used as food supplements to maintain and promote good health. Although several studies exist on the medicinal applications of *P. grandiflorum*, some people only use its roots to prepare food, tea, fermented beverages, or honey-based products. Using acetic acid bacteria, such as *Acetobacter aceti, Acetobacter pasteurianus*, and *Gluconobacter oxydans*, as a starter culture helps increase the production of acetic acid, thereby reducing the fermentation period when producing fermented vinegar [3,15,16].

The aim of this study was to investigate whether it could be used to prepare fermented vinegar without removing peel of bellflower roots (approximately 34% of the total root volume) in order to reduce peel waste. The secondary aims were to assess the quality of vinegar produced from peeled and unpeeled roots by analyzing their metabolite and volatile compound profiles as well as to compare their bioactive effects.

## 2. Materials and Methods

### 2.1. Preparation of Bellflower Root Samples

Bellflower roots of *P. grandiflorum* (L.) were purchased from a local market in Gyeonggi, Korea. After being washed, half of the roots were peeled to separate the edible pulp, and the other half were left unpeeled. All roots were heat-treated for two to three days in a dry oven at 40 °C, and then 1 kg samples were stored in a freezer at −20 °C prior to use. The unused part of the roots, including root waste, such as peels, rootlets, and heads, constituted approximately 34.29 ± 3.48% of the bellflower samples [17].

### 2.2. Preparation of Starter Culture

In this study, *Acetobacter pasteurianus* A11-2 (KACC 92203P) was used because it has the highest acidogenic ability of all traditional vinegar isolates and is registered at the Korean Agricultural Culture Collection (KACC, Wanju, Korea). This strain was isolated from brown rice vinegar produced into local farm at Kangwon province on 2017. Briefly, *A. pasteurianus* A11-2 were inoculated into an acetobacter solid medium containing a yeast extract (0.5%), glucose (3.0%), CaCO_3_ (1.0%), EtOH (3.0%), and agar (2.0%) and then incubated at 30 °C for 72 h until a clear zone was visible. This was then followed by subculturing in a maintenance broth medium containing a yeast extract (0.5%), glucose (0.5%), glycerin (1.0%), MgSO_4_·7H_2_O (0.02%), EtOH (6.0%), and acetic acid (1.0%) and then cultivation in a shaking incubator (JSSI-100; JS Research, Gongju, Korea) at 30 °C and 180 rpm for 72 h [18]. And alcoholic beverage used in the study was *Yakju* (Koran rice wine), which is fermented for 90 days with rice and glutinous rice charged 45% conc. and filtered, to purchase from local farm located at Cheongju city.

The vinegar starter culture was prepared by mixing 10 mL of an acetic bacterial suspension (total acidity: 4–5%, *v*/*v*) that was cultured from the abovementioned maintenance broth with 90 mL of a previously fermented alcoholic beverage (14.06% alcohol, pH 4.05, 0.43% acidity) with an alcohol concentration adjusted at 6–8% (pH 3.87, 0.28% acidity within alcohol and water mixtures). This mixture was further mixed with 900 mL of alcohol adjusted at 6–8% conc. and fermented at 30 °C to increase the starter culture amount to 1 L: this cultured vinegar starter was used to produce bellflower vinegar (BV).

### 2.3. Vinegar Production Using Bellflower Roots

Figure 1 depict the processes of manufacturing vinegar from peeled (BV2, BV4) and unpeeled (BV1, BV3) bellflower roots. For acetic fermentation, two root samples were first prepared by mixing 80 g of bellflower roots with 920 mL of an alcoholic beverage containing boiled water (alcohol conc. 7.18%, pH 3.87, 0.28% acidity) and then placed in a 1000 mL sterile glass vessel. Then, one culture (BV3, BV4) was inoculated with the starter culture including *A. pasteurianus* A11-2 of 2.5 × 10^7^ cfu/mL, whereas the other (BV1, BV2) was not (control). All cultures were incubated at 30 °C for 15 days.

### 2.4. Analysis of pH, Titratable Acidity, Ethanol, and Acetic Acid

The pH of the vinegar samples was determined using a pH meter (Orion Star A211; Thermo Fisher Scientific, Waltham, MA, USA). The titratable acidity, expressed as acetic acid content, was determined by titrating 1 mL of the sample with 0.1 N NaOH until the pH reached 8.25–8.30 [19].

The ethanol content was measured in the alcoholic beverage and vinegar according to the method of Kharchoufi et al. [20] using distillation equipment (Vapodest 200; Gerhardt GmbH, Königswinter, Germany) and a density meter (DMA 5000 M; Anton Paar GmbH, Wundschuh, Austria).

The acetic acid content in BV was analyzed by high-performance liquid chromatography (HPLC, Shimadzu Corp., Kyoto, Japan) using samples previously filtrated using a 0.45 μm membrane (Sigma-Aldrich, St. Louis, MO, USA). The filtrate was then injected into a YMC-Triart C18 column (150 × 3.00 mm i.d., S-3 μm; YMC Co., Ltd., Kyoto, Japan). The retention time and peak area of the standard were used to calculate the acetic acid content of the fermentation broth. The chromatographic conditions, that is, column temperature, mobile phase, flow rate, injection volume, and wavelength of detection, were 40 °C, 0.1% phosphoric acid, 0.4 mL/min, 2 μL, and UV 210 nm, respectively.

### 2.5. Gas Chromatography–Mass Spectrometry–Based Metabolite Analysis

To analyze the volatile and nonvolatile metabolites, we used a method described by Kim et al. [21] with some modifications. Briefly, 500 μL of vinegar diluted 200 times with distilled water was saturated with NaCl. Then, 1.5 mL of distilled water was added and the mixture was placed in headspace solid-phase microextraction (HS-SPME) vials (20 mL). 2-Methyl-1-pentanol (0.01 μL/L) was used as an internal standard. Vials were equilibrated at 25 °C for 15 min with stirring.

Volatile metabolites in the headspace were first absorbed by a fiber-coated SPME 50/30 μm assembly DVB/CAR/PDMS (StableFlex, 2 cm; Supelco Inc., Bellefonte, PA, USA) at 25 °C for 10 min and then fiber-injected into a gas chromatography–mass spectrometry (GC-MS) apparatus.

To analyze the metabolites, 3 μL of vinegar was completely dried using a CentriVap vacuum concentrator (Labconco, Kansas City, MO, USA) and processed via silylation using methoxyamine hydrochloride and *N*,*O*-bis(trimethylsilyl)trifluoroacetamide (BSTFA). Dichlorohexyl phthalate was used as an internal standard and added with pyridine and methoxyamine hydrochloride, and the reaction was performed at 37 °C for 90 min. After cooling down to room temperature, BSTFA was added to derivatize the extract at 70 °C for 30 min and then transferred to a vial. The derivatized metabolites were analyzed by GC-MS using a GC-2010 Plus apparatus (Shimadzu, Tokyo, Japan) coupled to a GCMS-TQ8030 apparatus (Shimadzu). To analyze the volatile and derivatized nonvolatile vinegar metabolites, we used DB-5 and DB-WAX columns (30 m × 0.25 mm i.d., 0.25 μm film thickness; J & W Scientific, Santa Clara, CA, USA). The volatile metabolites were absorbed onto SPME, and a 1 μL aliquot of derivatized samples was injected into the column with a split ratio of 1: 10 and 1: 50, respectively. Helium was used as the column carrier gas at a flow rate of 1 mL/min. For the volatile metabolites, the injector temperature was set at 250 °C and the oven temperature was held at 40 °C for 3 min, increased to 90 °C at a rate of 5 °C/min, then increased to 230 °C at a rate of 19 °C/min, and maintained at 230 °C for 5 min. On the other hand, for the nonvolatile metabolites, the injector temperature was set at 200 °C and the oven temperature was held at 70 °C for 2 min, increased to 210 °C at a rate of 7 °C/min, then increased to 320 °C at a rate of 10 °C/min, and maintained at 320 °C for 7 min. The GC column effluent was analyzed in the electron impact ionization mode. The temperatures of the ion source and the interface were 230 °C and 250 °C, respectively. The ions were generated at 15 eV, and the effluents were analyzed within a scan range of 45–550 *m*/*z* with a scan event time of 0.3 s and scan speed of 2000 u/s. The detector voltage was 0.1 kV using a 100 threshold.

### 2.6. Antibacterial Activity

Four pathogenic strains were purchased from the Korean Collection for Type Cultures (KCTC, KACC, and ATCC (Virginia, USA): *Escherichia coli* KCTC 1309, *Salmonella* Typhimurium KCTC 41028, *Bacillus cereus* KACC 10004, and *Staphylococcus aureus subsp. aureus* ATCC 6538. All strains were grown in tryptic soy broth (Difco, Detroit, MI, USA) and incubated overnight at 30 °C for 24 h before use. All inocula were adjusted to an optical density of 0.5 at 660 nm. A 100 μL aliquot of each strain was evenly swabbed onto a tryptic soy agar plate that was previously dried and stored at 37 °C, and then 8 mm paper discs (Advantec, Toyo Roshi Kaisha Ltd., Tokyo, Japan) were placed onto the plates. Then, 50 μL of each BV from different fermentation durations was added to the paper discs. Plates were incubated at 30 °C for 18 h, and the inhibition zones surrounding the samples were measured. All experiments were repeated at least three times.

### 2.7. Cell Culture of RAW 264.7 Mouse Macrophage Cells

RAW 264.7 mouse macrophage cells were used in this study (KCLB 40071; Korean Cell Line Bank, Seoul, Korea). Cells were maintained in Dulbecco’s modified Eagle’s medium (Invitrogen, Carlsbad, CA, USA) supplemented with 10% fetal bovine serum (Invitrogen) and 1% Pen/Strep (Invitrogen) at 37 °C in the presence of 5% CO_2_. For all experiments, cells were grown beyond 80% confluence with a maximum of five passages. Cells were then cultured in 96-well microplates (5 × 10^5^ cells/well) and treated with 0.01%, 0.05%, and 0.1% (*v*/*v*) vinegar, which has been fermented for 10 days, for 24 h. Untreated cells with no added vinegar were used as the control. Cell proliferation was assessed using a 3-(4,5-dimethylthiazol-2-yl)-2,5-diphenyltetrazolium bromide (MTT) assay [22] and calculated as follows: (absorbance of sample/absorbance of control) × 100.

### 2.8. Analysis of the Levels of Cytokines and Chemokines

Macrophage cells were cultured for 24 h and then treated for 4 h with vinegar, which has been fermented for 10 days, and 100 ng/mL lipopolysaccharides (LPS, Sigma-Aldrich, St. Louis, MO, USA) to activate the cells. Cultures were then incubated at 37 °C in 5% CO_2_ (VS-9160C; Vision Scientific Co., Ltd., Daejeon, Korea). After 20 h, the supernatant was collected to analyze the cytokine level using a Proteome Profiler Mouse Cytokine Array Kit (ARY006; R & D Systems, Minneapolis, MN, USA). Each sample was then diluted with a cocktail of biotinylated antibodies on a mouse cytokine array membrane, to retain cytokine–antibody complexes in the membrane, and then incubated.

After washing three times, streptavidin-horseradish peroxidase and chemiluminescent detection reagents were sequentially added. Array images were obtained using an Amersham Imager 680 image analyzer (GE Healthcare, Chicago, IL, USA) after 6–7 min of exposure and analyzed by densitometry for integral optical density (IOD) using Quick Spots (version 22.0.1b; Western Vision Software, Salt Lake, UT, USA). The IOD of each cytokine was normalized to the corresponding negative control IOD results.

### 2.9. Statistical Analysis

All experiments were performed in triplicate. Values are expressed as the mean ± standard deviation (SD) and processed by SAS (SAS Institute, Cary, NC, USA). A *p*-value of <0.05 was considered statistically significant. Comparisons were made using one-way analysis of variance (ANOVA) and Duncan’s multiple range test. An independent *t*-test was used to examine cell viability and cytokine levels. A statistical heatmap analysis and a principal component analysis (PCA) were performed using XLSTAT for Microsoft Excel (Microsoft Corp., Redmond, WA, USA).

## 3. Results and Discussion

The aim of this study was to develop cost-effective processing techniques to obtain vinegar from unpeeled bellflower roots, thus reducing waste and adding value to the product. Bellflower root peels have no value as a food ingredient, and unpeeling the roots is a labor-intensive and noneconomical task. Moreover, peels have an unpleasant bitter taste due to the presence of metabolites like saponins and tannins. Nevertheless, they contain other bioactive components, such as platycodin D. Besides their economic benefits, many consumers expect higher functional or healthy foods made from recycled waste.

### 3.1. Analysis of pH, Titratable Acidity, and Ethanol

To produce vinegar from peeled and unpeeled bellflower roots using *Acetobacter pasteurianus*, acetic fermentation was performed at 30 °C for 15 days (see Figure 1). Here, we defined BV1-BV4 vinegar samples as follows: BV1, vinegar from unpeeled bellflower root; BV2, vinegar from peeled bellflower root; BV3, vinegar from unpeeled bellflower root using *A. pasteurianus* A11-2; BV4, vinegar from peeled bellflower root using *A. pasteurianus* A11-2. The initial pH of all BV mixtures (BV1–BV4) containing an alcoholic beverage and a starter culture was in the range 3.85–4.05, which decreased to 3.15–3.20 after 15 days of fermentation. Notably, BV3 reached the lowest pH (3.18) on day 10. Using a batch-type acetifier, Singh and Singh [23] showed that the pH of wine vinegar decreases from 4.50 to 3.50 on day 9. Roda et al. [3], on the other hand, observed that the pH decreased from 3.78 to 3.00 during acetic fermentation over a period of 30 days.

In this study, the acidity of bellflower root vinegar increased from 0.23–0.37% on the initial day to 5.41% and 5.59% on day 10 (BV3) and day 15 (BV1), respectively, as indicated in Table 1. These results are comparable to those of Roda et al. [3], who obtained an increase of 0.05–5.00% (w/v) in the production of acetic acid from pineapples over 30 days of fermentation. In the vinegar sample with starter culture (BV3), the acidity increased dramatically over 10 days but eventually decreased to 4.46% at the end of the acetification process. This phenomenon could be due to the peroxidation of acetic acid into carbon oxide [24].

Ethanol is the key substrate for acetic fermentation, whose concentration might decrease from 6–7% (initial) to 0% by the end of fermentation. In this study, the initial concentrations of ethanol in BV3 and BV4 were 6.12% and 5.84%, respectively, which were not detected after 10 days of fermentation. Furthermore, no ethanol was detected in vinegar on day 15, regardless of the presence of a starter culture. Acetic acid bacteria are used as a starter culture because they mediate oxidation, which increases the concentration of acetaldehyde at the beginning of the acetification process [25].

In our study, the fermentation period to produce vinegar was shortened by using the vinegar starter culture, it was confirmed that the acidity and acetic acid was higher than that of the peeled bellflower from vinegar using unpeeled bellflower.

### 3.2. Metabolite Profiles of Bellflower Root Vinegar Analyzed by GC-MS

Conversion of ethanol into acetic acid by acetic acid bacteria is considered the main process in vinegar fermentation. However, fermentation may also lead to the synthesis or breakdown of compounds with sensory properties (sugars, amino acids, organic acids, alcohols, etc.) depending on the fermentation conditions, such as temperature, starter culture, and incubation time. Nontargeted quadrupole time-of-flight mass spectrometry analysis revealed a significant change in metabolite composition before and after fermentation for 15 days using PCA (Figure 2A). The reason why the metabolite profile was determined was to find the necessary parameters for monitoring and controlling quality during fermentation [26].

Additionally, the concentration of organic acids, such as malic acid, acetic acid, gluconic acid, and citric acid, was found to be significantly high in samples BV3 and BV4, regardless of whether peeling was performed or whether a starter culture was used (Figure 2A). These compounds are produced by *A. pasteurianus* under the influence of chemical and enzymatic changes, as reported by Roda et al. [3]. Notably, the most prominent change observed in the metabolite profile was the decrease in amino acids and sugars after 10 days of fermentation, unlike the observed changes in organic acids. Variations in the amino acid composition, such as L-proline and L-leucine, which are primary sources of nitrogen, during vinegar fermentation have been attributed to cell lysis [27]. This is in line with what has been mentioned by Roda et al. [3], who found evidence that the content of amino acids decreases with the growth of acetic acid bacteria. According to previous reports on *P. grandiflorum* [10], the BV samples contained several amino acids and fatty acids with bioactive components, including saponins and flavonoids, which are consumed during fermentation. Overall, the profile of BV2 remained unchanged, although its 4-aminobutanoic acid, myo-inositol, and sorbitol content was different from that of other vinegar samples. This can be explained by the abnormal fermentation due to the antibacterial effect of peeled roots against wild-type acetic acid bacteria, as the content of bioactive compounds in *P. grandiflorum* is higher in peeled roots than in unpeeled roots. As previously described, brewed vinegar is rich in essential amino acids, such as alanine, glutamic acid, lysine, and leucine in rice vinegar [28] and alanine in apple vinegar [29]. In our results, however, above described amino acids were not highly showed during vinegar fermentation for 15 days.

### 3.3. Profile of Aromatic Volatiles in Bellflower Root Vinegar

The flavor components of fermented BV were analyzed using HS-SPME and GC-MS. Additionally, the aroma of different types of vinegar was studied using GC-MS to identify the compounds responsible for the aromatic notes associated with the selected descriptors from the typical aroma, as reported by Callejón et al. [30].

The main flavor volatiles in all BV samples fermented for 10–15 days were acetic acid, ethyl ester, butyl formate, isoamyl acetate (1-butanol, 3-methyl-, acetate), nonanal, methyl salicylate, and phenethyl acetate. These compounds are known to impart fruity or floral odors (Table 2), whereas the odor properties of methyl 2-hydroxypropanoate and malonic acid are unknown. On the contrary, 2,4,5-trimethyl-1,3-dioxolane, 1,5-pentanediol, 1-pentanol, and benzaldehyde, which are found in alcoholic beverages, were detected in the BV samples during initial fermentation. Chung et al. [31] reported a total of 103 components from bellflower roots identified using GC-MS, including main components such as 1-hexanal, *trans*-2-hexenal, 1-hexanol, *cis*-3-hexenol, *trans*-2-hexenol, and 1-octen-3-ol, perceived by sensory evaluation as a grass-like odor, from bellflower roots. In agreement with their results, we detected 1-hexanol and ethyl acetate in the BV samples at the beginning and after fermentation.

The aromatic profile results showed the same difference for some compounds, e.g., ethyl acetate and isoamyl acetate. Especially, ethyl acetate in BV1 and BV2 was higher at the value of 16.24 and 17.26 μg/mL than that in BV3 and BV4 at 1.16 and 0.31 μg/mL. Ethyl acetate, which is formed between acetic acid and ethanol, was high in BV1 and BV2 without starter bacteria compared with BV3 or BV4 with vinegar starter. Herein, we found that the component was naturally formed between residual ethanol and the produced acetic acid during fermentation for 15 days. The important role of vinegar starter to improve the production of acetic acid in order to shorten the fermentation time is known.

### 3.4. Antibacterial Activity

We assessed the antibacterial effect of BV on *E. coli*, *S.* Typhimurium, *B. cereus*, and *S. aureus* by the disk-diffusion agar method, using aliquots from different fermentation durations. As shown in Figure 3, maximum inhibition was observed with undiluted vinegar samples (approximately 5% acidity) fermented for 10 days. No inhibition effect was observed in any of the BV samples before five days of fermentation. A similar inhibition period, from day 10 to day 15, was observed for all BV samples. This result suggested that the high inhibition against pathogenic bacteria after 10 days of fermentation may have been caused by an increase in acidity. Furthermore, among all BV samples, BV3 showed the highest inhibitory effect on all tested bacteria. 

In contrast, *B. cereus* showed the least inhibition susceptibility among all bacterial species. Unlike the other three pathogens in our research, *B. cereus* are sporulated bacteria whose spores and capsules are important for their survival, because they increase their resistance against adverse environmental conditions, such as nonoptimal pH and temperature and antibacterial agents.

It has also been reported that the antibacterial effect of vinegar is due to the presence of organic acids, such as acetic acid and lactic acid. Moreover, bellflower roots are also known to contain several phenolic compounds, such as flavonoids, polyphenols, and saponins (e.g., platycodin) [32]. According to Medina et al. [33], compared with aqueous extracts of wine, olive oil, and other beverages, vinegar has the strongest inhibitory effect against all tested pathogens. These results are in line with our study, as acetic acid has a stronger effect at a concentration below 3% than that of antiseptics used as a control [34]. Therefore, it was confirmed that the high antibacterial activity of vinegar by using vinegar starter culture and unpeeled bellflower roots was caused by various useful substances and high acidity. On the basis of these results, in the future, we will focus on investigating the properties of BV using the minimum inhibitory concentration.

### 3.5. Effect of BV on Modulating Cytokine and Chemokine Expression in RAW 264.7 Cells

Macrophages are components of the innate immune system that are responsible for the phagocytosis of bacteria, damaged cells, and unknown tissues. Upon phagocytosis, macrophages produce cytokines, chemokines, and other metabolites.

As reported by Arango Duque and Descoteaux [35], cytokines and chemokines are low-molecular-weight proteins that mediate intercellular communication and are secreted by different cell types, primarily those of the immune system. These proteins act as potent signaling molecules to regulate inflammation and play a crucial role in modulating the immune system. Classical (M1) macrophage activation is induced by interferon-gamma or LPS and promotes a proinflammatory response, whereas alternative (M2) macrophage activation is induced by interleukin-4 (IL-4), IL-10, or IL-13 and stimulates an anti-inflammatory response.

To the best of our knowledge, this is the first study evaluating the immune-response-enhancing effects of bellflower root vinegar. As a first step, we performed a cytotoxicity assay to assess the effect of BV on the viability of RAW 264.7 cells (data not shown). We used BV concentrations ranging from 0.01% to 0.1%, as no cytotoxic effect on RAW 264.7 cells was observed within this range, and obtained a cell viability at 0.01% of BV3 and BV4 of over 111.7% and 151.4%, respectively, compared with the untreated control group.

To examine the immune-enhancing effect of BV, we stimulated RAW 264.7 macrophage cells with LPS and 0.01% of each BV. We also measured the expression of several cytokines, including the interleukin family (e.g., IL-1, IL-6, IL-10, and IL-12) and tumor necrosis factor-α (TNF-α), and chemokines, including macrophage inflammatory proteins (MIPs), such as MIP-1α, MIP-1β, and MIP-2, as well as regulated on activation, normal T-cell expressed and secreted (RANTES) (Figure 4), using the Proteome Profiler Mouse Cytokine Array Kit (R & D Systems, Bio-Techne). The most prominent effect observed in the immunomodulation of macrophage cells was the significant increase in IL-1α and TNF-α, which are known to be secreted by macrophages upon stimulation by LPS. In addition, BV3 caused a higher expression of cytokines than BV4 did and significantly increased the production of IL-1α and TNF-α in RAW mouse macrophage cells, cytokines that have been reported to have antitumor properties. Previous reports have described that several Th1-promoting cytokines, such as TNF-α and IL-12, can enhance antitumor immunity [36,37]. Moreover, it has been shown that several chemokines, including MIP-1α, MIP-1β, MIP-2, and RANTES, are upregulated. For instance, the expression of MIP-1α, MIP-1β, and MIP-2 in BV3-treated RAW cells increased 2.5-, 0.6-, and 0.4-fold, respectively (Figure 4), compared with the control cells (*p* < 0.05, *n* = 3; Figure 4B). Moreover, MIP-1α and MIP-1β are known to regulate the expression of chemotactic chemokines secreted by macrophages.

In conclusion, all of these results suggest that bellflower root vinegar from unpeeled roots fermented with *A. pasteurianus* can activate macrophage cells and upregulate the expression of cytokines and chemokines in vitro. However, further studies are required to evaluate the immune-response-enhancing effects of bellflower root vinegar in animal models before commercialization.

## 4. Conclusions

*Platycodon grandiflorum* is a well-known herb used as a medicinal ingredient and healthy type of food thanks to its content of amino acids and saponins, including platycodin D, as bioactive substances. Most types of fermented vinegar are made from nutrient-rich grains and fruits. However, vinegar produced from herbs has rarely been reported. Therefore, our results are considered to be noteworthy as they show the metabolic and functional differences among different types of vinegar obtained from peeled and unpeeled bellflower roots fermented with *Acetobacter pasteurianus*. We also demonstrated that fermented vinegar from bellflower unpeeled roots, with an acidity of 5.59% (w/v), can be manufactured using *A. pasteurianus* A11-2 in ceramic vessels over 10 days at 30 °C. Notably, the vinegar produced under these conditions showed favorable antibacterial properties against four pathogens, concomitant with the levels of acetic acid, as well as immune-response-enhancing effects in mouse macrophages.

Therefore, in conclusion, we propose that bellflower root peel waste can be effectively used in manufacturing fermented vinegar to be used as a medicinal and healthy food product.

## Figures and Tables

**Figure 1 foods-09-01063-f001:**
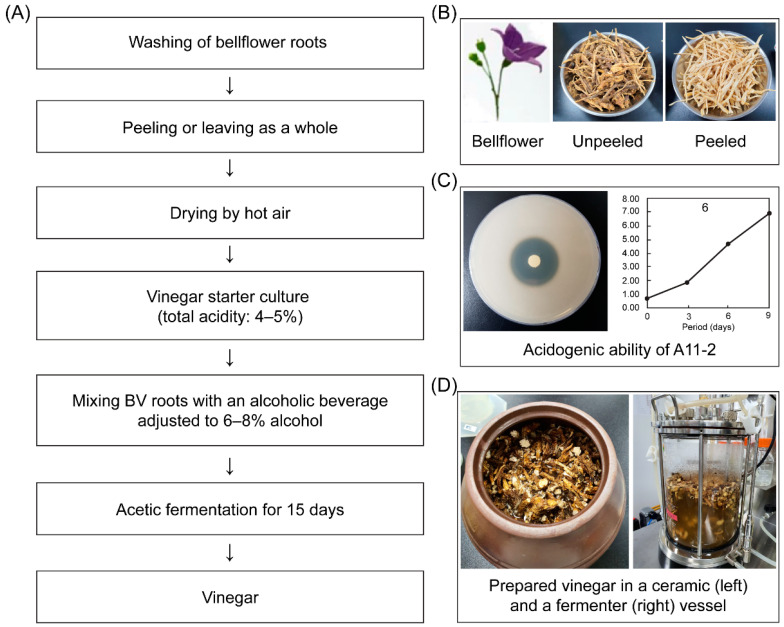
Flow chart of vinegar manufacturing using peeled and unpeeled bellflower roots. (**A**) Processing of bellflower vinegar (BV). (**B**) Raw material of bellflower. (**C**) Acidogenic ability of *Acetobacter pasteurianus* A11-2 as a starter culture. (**D**) Fermented vessels (traditional ceramic and fermenter) used for vinegar fermentation.

**Figure 2 foods-09-01063-f002:**
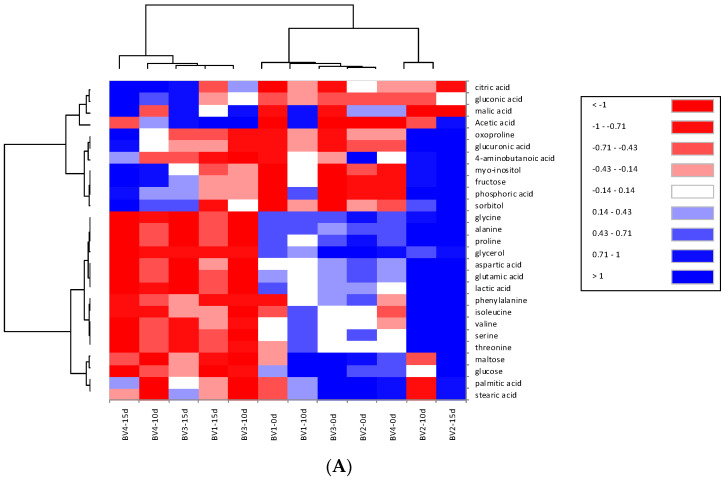
Heatmap and principal component analysis (PCA) of metabolites and aromatic volatile compounds outlining bellflower root vinegar during fermentation. (**A**) metabolites, (**B**) aromatic volatile compounds, (**C**,**D**), expressed by PCA analysis.

**Figure 3 foods-09-01063-f003:**
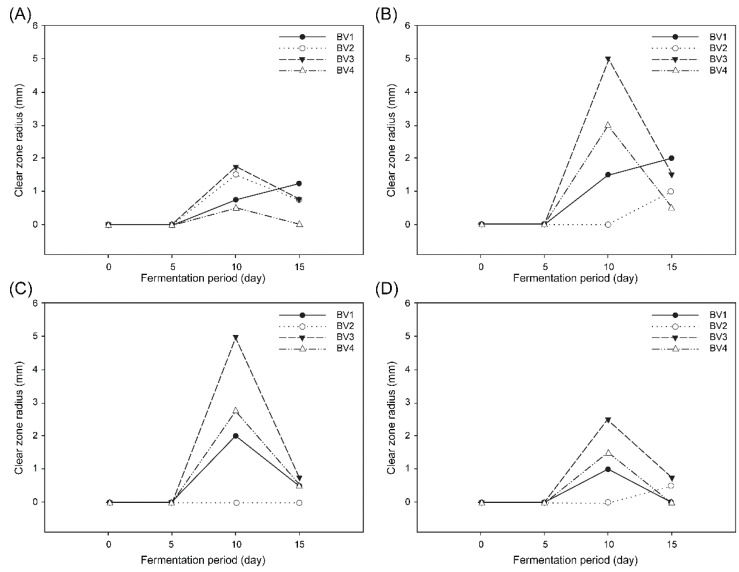
Inhibitory effect of bellflower root vinegar on bacterial strains. (**A**) *Bacillus cereus*, (**B**) *Staphylococcus aureus*, (**C**) *Escherichia coli*, and (**D**) *Salmonella* Typhimurium. BV1, vinegar from unpeeled bellflower root; BV2, vinegar from peeled bellflower root; BV3, vinegar from unpeeled bellflower root using *Acetobacter pasteurianus* A11-2; BV4, vinegar from peeled bellflower root using *Acetobacter pasteurianus* A11-2.

**Figure 4 foods-09-01063-f004:**
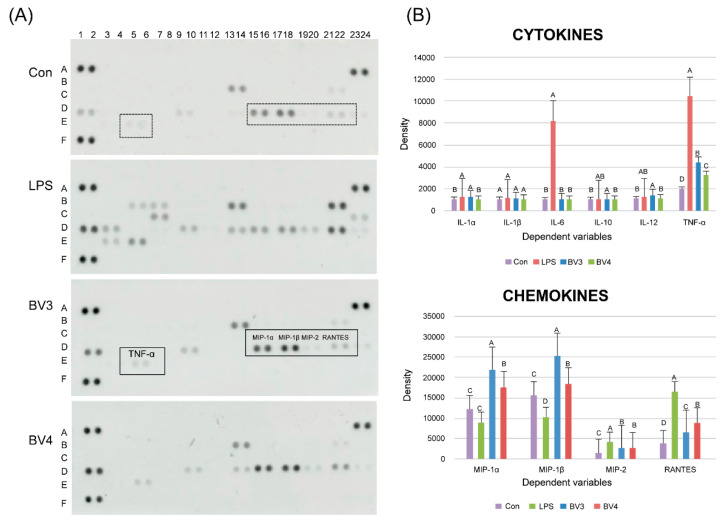
Effect of BV samples on the production of immune-enhancing cytokines and chemokines in RAW 264.7 macrophages. (**A**) Cytokine and chemokine array. (**B**) Densities of cytokines and chemokines by image analysis. Con, control without BV samples and LPS; BV3, with unpeeled roots and a starter culture; BV4, with peeled roots and a starter culture.

**Table 1 foods-09-01063-t001:** Changes in pH, titratable acidity (%), alcohol (%), and content of acetic acid in vinegar produced from fermented bellflower roots. ^(1)^ BV1, vinegar from unpeeled bellflower root; BV2, vinegar from peeled bellflower root; BV3, vinegar from unpeeled bellflower root using *Acetobacter pasteurianus* A11-2; BV4, vinegar from peeled bellflower root using *Acetobacter pasteurianus* A11-2. ^(2)^ Results from the same fermentation time (A–D) or group (a–d) are significantly different (*p* < 0.05) as per Duncan’s multiple range test.

Group ^(1)^	Period (Days)	pH	Titratable Acidity (%)	Alcohol (%)	Acetic Acid (ppm)
BV1	0	3.96 ± 0.05 ^(2),B,b^	0.23 ± 0.00 ^C,b^	6.88 ± 0.00 ^A,a^	4.07 ± 0.31
	5	4.18 ± 0.01 ^A,a^	0.29 ± 0.01 ^C,b^	5.76 ± 0.17 ^A,B,a^	7.41 ± 1.07
	10	3.37 ± 0.04 ^C,b^	3.08 ± 0.66 ^B,b,c^	3.53 ± 1.66 ^B,a^	47.44 ± 1.93
	15	3.15 ± 0.06 ^D,a^	5.41 ± 0.53 ^A,a^	0.00 ± 0.00 ^C,a^	53.32 ± 4.94
BV2	0	4.05 ± 0.04 ^A,a^	0.24 ± 0.00 ^B,b^	6.88 ± 0.11 ^A,a^	3.94 ± 0.37
	5	4.20 ± 0.00 ^A,a^	0.32 ± 0.00 ^B,b^	5.76 ± 0.06 ^B,a^	3.89 ± 0.16
	10	3.60 ± 0.13 ^B,a^	1.18 ± 0.80 ^B,c^	0.20 ± 0.07 ^C,b^	15.67 ± 5.47
	15	3.18 ± 0.01 ^C,a^	4.80 ± 0.37 ^A,a^	0.00 ± 0.00 ^C,a^	49.24 ± 4.35
BV3	0	3.85 ± 0.00 ^A,c^	0.36 ± 0.00 ^B,a^	6.12 ± 0.17 ^A,b^	6.43 ± 0.31
	5	3.77 ± 0.09 ^A,b^	1.11 ± 0.31 ^B,a^	3.74 ± 0.82 ^B,b^	15.07 ± 1.83
	10	3.18 ± 0.03 ^B,b^	5.59 ± 0.96 ^A,a^	0.00 ± 0.00 ^C,b^	59.85 ± 0.97
	15	3.19 ± 0.05 ^B,a^	4.46 ± 1.99 ^A,a^	0.00 ± 0.00 ^C,a^	49.76 ± 20.10
BV4	0	3.95 ± 0.01 ^A,b^	0.37 ± 0.00 ^B,a^	5.84 ± 0.11 ^A,b^	6.49 ± 0.31
	5	4.09 ± 0.01 ^A,a^	0.42 ± 0.03 ^B,b^	3.98 ± 0.65 ^B,b^	4.86 ± 0.56
	10	3.31 ± 0.09 ^B,b^	3.56 ± 0.27 ^A,b^	0.00 ± 0.00 ^C,b^	35.18 ± 1.06
	15	3.20 ± 0.11 ^B,a^	2.11 ± 1.08 ^A,a^	0.00 ± 0.00 ^C,a^	18.50 ± 17.75

**Table 2 foods-09-01063-t002:** Odor description of volatiles found in bellflower vinegar (BV) samples by gas chromatography–mass spectrometry (GC-MS).

Compound	Synonyms	Odor Description	BV Samples on Day 15			
			BV1	BV2	BV3	BV4
*ESTERS*						
Acetic acid ethyl ester	Ethyl acetate	ethereal, fruity	16.24	17.26	1.16	0.31
Benzeneethanol	2-Phenylethanol	pleasant floral	0.74	0.76	0.51	0.37
3-Methyl-1-butanol acetate	Isoamyl acetate	juicy fruit	1.64	1.42	0.86	0.21
Butanedioic acid, diethyl ester	Diethyl succinate	fruity	0.03	0.03	0.02	0.03
Butyl formate	*n*-Butyl formate	fruity	0.53	0.28	0.30	0.09
Ethyl hexanoate	Ethyl caproate	fruity	0.00	0.00	0.00	0.00
Ethyl 2-hydroxypropanoate	Ethyl lactate	creamy, fruit	0.03	0.46	0.08	0.00
Ethyl-hydroxycaproate	-	unknown	0.01	0.02	0.00	0.00
Ethyl benzoate	Benzoic acid, ethyl ester	pleasant fruity	0.02	0.04	0.02	0.02
Methyl-2-hydroxypropanoate	Methyl lactate	unknown	0.65	0.79	0.45	0.40
Methyl salicylate	-	mint	1.69	1.53	1.69	1.05
Phenethyl acetate	-	floral, sweet honey	0.68	0.72	0.41	0.15
*ALCOHOLS*						
1-Hexanol	Hexyl alcohol	fresh	0.00	0.30	0.23	0.00
1,5-Pentanediol	Pentane-1,5-diol	characteristic odor	0.10	0.03	0.08	0.02
1-Pentanol	-	characteristic strong	3.08	1.55	2.33	0.67
2-Ethyl-1-hexanol	2-Ethylhexanol	aromatic	0.37	0.32	0.34	0.41
*OTHERS*						
Hydroxypropanoic acid	-	unknown	2.08	2.44	0.07	0.01
malonic acid	Carboxyacetic acid	unknown	8.96	7.42	11.25	3.09
Benzaldehyde	-	almond-like	0.05	0.03	0.70	0.35
1,3-Di-*tert*-butylbenzene	1,3-Bis(1,1-dimethylethyl)benzene	unknown	0.03	0.05	0.04	0.04
Nonanal	-	citrus	0.00	0.00	0.00	0.00
2,4,5-Trimethyl-1,3-dioxolane		alcohol	0.34	0.16	0.37	0.03

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
