# Peer review of "Metabolite Profile and Immunomodulatory Properties of Bellflower Root Vinegar Produced Using *Acetobacter pasteurianus* A11-2"

_foods, 2020, doi:10.3390/foods9081063_

Round 1

Reviewer 1 Report

General comments:

The work deals with the metabolite profile and immunomodulatory properties of bellflower root vinegar 1 produced using Acetobacter pasteurianus A11-2.

The manuscript is well-written and based upon good laboratory research that was supported by an appropriate statistical analysis. Although this study can be considered as a new and original contribution to the agricultural and food science, there are few points that should be considered to improve the scientific merit of the manuscript.

Specific comments:

  1. MATERIALS AND METHODS: This section lacks a subsection entitled “Strains” to explain the origin of the different strains used in the manuscript.

2.2. Subsection: Preparation of starter culture. Lines 102-108.

2.1.1. How was the fermented alcoholic beverage obtained?

2.1.2. The author said: “The vinegar starter culture was prepared by mixing 10 mL of an acetic bacterial suspension (total acidity: 4-5%, v/v) that was cultured from the abovementioned maintenance broth with 90 mL of a previously fermented alcoholic beverage (11.06% alcohol, pH 4.05, 0.43% acidity) with an alcohol concentration adjusted at 6–8% (pH 3.87, 0.28% acidity within alcohol and water mixtures). This mixture was further mixed with 900 mL of 30% alcohol to increase the starter culture amount to 1 L: this cultured vinegar starter was used to produce bellflower vinegar (BV). It is not clear why did the authors reduce the alcohol concentration of the fermented alcoholic beverage at 6–8% and then mixed the mixture of this fermented beverage and the acetic bacterial suspension with 900 mL of 30% alcohol. The latter process increased the final ethanol concentration in the starter culture.

2.3. Subsection: Analysis of pH, titratable acidity, ethanol, and acetic acid. Lines 129-132.

2.3.1. Why was the ethanol content measured with the method of Kharchoufi et al. (2018) and not by high-performance liquid chromatography (HPLC)? The latter method is much more precise than the first one. In fact, the acetic acid content was measures by HPLC.

2.6. Subsection: Antimicrobial activity. Lines 181-192.

2.6.1. Line 181. The name of this subsection is incorrect because the authors determine the inhibitory activity of each fermented BV against four bacteria: Escherichia coli KCTC 1309, Salmonella typhimurium KCTC 41028, Bacillus cereus KACC 10004, and Staphylococcus aureus subsp. aureus ATCC 6538. Therefore, the authors determine the antibacterial activity of each fermented BV rather than their antimicrobial activity. In consequence, the correct name of this subsection should be Antibacterial activity.

  1. RESULTS AND DISCUSSION

This section should be summarized to give and discuss in depth the results obtained by the authors.

Author Response

Dear reviewer,

Thank you so much for your time and efforts to come up with these valuable comments to improve our manuscript quality. The attached file are our answers
to your concerns.

Reviewer 2 Report

The work by Na-Young GIL described the production of a vinegar from unpeeled roots of bellflowers (Platycodon grandiflorum) using Acetobacter pasteurianus A11-2, and compared it with vinegar obtained from peeled roots and with wild bacteria. The work is interesting for readers in food processing and functional food. Although the aim of the work is not clearly stated and discussed in the manuscript. The differences found in the vinegar characterization of the 4 different samples tested are not clearly stated in all sections of the manuscript. The differences found in the fermentation using unpeeled roots should be highlighted in the manuscript discussion as the main conclusion of the work is that “bellflower root peel waste can be effectively used in manufacturing fermented vinegar to be used as a medicinal and healthy food product.”

Detailed comments on the manuscript might be found bellow.

Line 75 “The aim of this study was to investigate whether the peel waste of bellflower roots (approximately 34% of the total root volume) can be used to prepare fermented vinegar.” This sentence gives the idea that the peel itself will be fermented. Nevertheless, in the work the peel was not fermented alone, instead unpeel roots were fermented. Please clarify the sentence.

Line 104. Please describe the composition and/or the methodology used to obtain the fermented alcoholic beverage

Line 113. Please describe the composition of the alcoholic beverage

Line 115. Please provide the number of cells of A. pasteurianus A11-2 used in the starter culture.

Line 135. Just a note. Filters with porosity 0.45 μm are used to separate fungi cells while 0.2  μm are more appropriated to bacteria (which are smaller).

Section 3.1. With the abbreviations used it is hard to clearly understand the effect of the peel and the bacteria in the fermentation. Please write a paragraph stressing these effects.

Section 3.2 It is hard to understand the explanation related with the amino acids content. It is stated that amino acid are primary sources of nitrogen, so it should decrease during fermentation, nevertheless, later it is explained that its no increase is related to the bacteria cell autolysis…Please check:  Line 296 “ amino acid composition, … are primary sources of nitrogen, during vinegar fermentation have been attributed to cell lysis” Please address also which kind of cells are you citing in each specific sentence.; and  Line 308 “Our results, however, showed no increase in the amino acid content during vinegar fermentation for 15 days, possibly due to bacterial cell autolysis after nitrogen intake by  acetic acid bacteria”

Section 3.3 The differences found in volatile compounds of the different fermented samples are not address or discussed in the section. Nevertheless, the profile was very different for some compounds, e.g. acetic acid obtained the following values: 16.24, 17.26, 1.16, 0.31

Section 3.4 Once more it is not discussed the differences between the samples assayed. Thus, results on peeled and unpeeled roots are not compared.

Fig.3 the meaning of BV1, 2,… should be written in the figure caption

Fig.4 Although BV1 and BV2 are cited in the figure caption, no results on these samples are provided.

Author Response

(The authors gave the same response as above.)
